rsos.royalsocietypublishing.org

Subject Areas:
electrical engineering/materials science

Keywords:
electrofluidic display, ParyleneC/AF1600X, thermal ageing, failure mode

Author for correspondence:
Biao Tang
e-mail: tangbiao@scnu.edu.cn

# Failure modes analysis of electrofluidic display under thermal ageing

Baoqin Dong[1,2], Biao Tang[1,2], Jan Groenewold[1,2,3], Hui Li[1,2], Rui Zhou[1,2], Alexander Victor Henzen[1,2] and Guofu Zhou[1,2,4,5]

[1]Guangdong Provincial Key Laboratory of Optical Information Materials and Technology and Institute of Electronic Paper Displays, South China Academy of Advanced Optoelectronics, South China Normal University, Guangzhou 510006, People's Republic of China
[2]National Center for International Research on Green Optoelectronics, South China Normal University, Guangzhou 510006, People's Republic of China
[3]Van 't Hoff Laboratory for Physical and Colloid Chemistry, Debye Research Institute, Utrecht University, Padualaan 8, 3584 CH Utrecht, The Netherlands
[4]Shenzhen Guohua Optoelectronics Tech. Co. Ltd, Shenzhen 518110, People's Republic of China
[5]Academy of Shenzhen Guohua Optoelectronics, Shenzhen 518110, People's Republic of China

BD, 0000-0002-6471-6871

Dielectric failure as well as optical switching failure in electrofluidic display (EFD) are still a bottleneck for sufficient device lifetime. In this study, a dielectric redundancy-designed multilayer insulator of ParyleneC/AF1600X was applied in an EFD device. The reliability performance was systematically studied by tracking the applied voltage-dependent leakage current and capacitance changes (I–V and C–V curves) with thermal ageing time. The multilayer insulator shows a more stable performance in leakage current compared to a single-layer insulator. The failure modes during operation underlying the single-layer and the multilayer dielectric appear to be different as exemplified by microscopic images. The single-layer AFX shows significant detachment. In addition, by quantitatively analysing the C–V curves with ageing time, we find that for the single AFX device, the dominant failure mode is 'no-opening' of the pixels. For the multilayer device, the dominant failure mode is 'no-closing' of the pixels. This study provides tools for distinguishing the basic failure modes of an EFD device and demonstrates a quantitative method for evaluating the reliability performance of the device under thermal ageing.

# 1. Introduction

Electrowetting-on-dielectric (EWOD) provides a way for reversible manipulation of surface wettability by externally applied voltage,

rsos.royalsocietypublishing.org R. Soc. open sci. 5: 181121

which inspires various applications in microfluidic devices, including micro-lens [1,2], lab-on-a-chip platform [3,4], heat transfer [5,6] and displays [7]. Among the above applications, electrofluidic display (EFD) has drawn much attention due to the high colour brightness, video speed, low power consumption and paper-like reflective display [8]. As a typical EWOD-based device with reversible-switching desire, reliability issues are one of the most critical challenges [9,10], which hinders the industrialization of these devices. Various failure modes, which cause device degradation that could be categorized into dielectric failure [11] and switching failure [12], have been reported and investigated. The physics behind these failure modes, like charge trapping [13,14] and leakage current [15,16], has been well discussed in many reviews [11,17]. Device performance can be improved by the enhancement of breakdown voltage [18,19] and dielectric strength [17,20]. In this context, various dielectric materials have been studied to improve the dielectric properties and stability [21–23]. Raj et al. explored the influence of different conducting liquids on electrowetting failure modes. This work suggests that large-sized ions in electrowetting liquids could eliminate charge trapping and improve reliability [24]. All the efforts have facilitated the development of EWOD application. In other research studies, multilayer insulators were introduced to show remarkable improvement in EWOD. By introducing a multilayer, optimization of materials can be achieved. The benefits of the materials can be combined, while the disadvantages can be reduced [25]. In a multilayer design in which a thin fluoropolymer layer is often coupled with an additional dielectric layer with high relative permittivity, the fluoropolymer is susceptible to break down due to an uneven distribution of electric fields. However, the effects of the fluoropolymer layer thickness on breakdown strength caused by the voltage intensity distribution were not well considered or designed [26]. As for fluoropolymer films, the water-tree [27] and mechanical breakdown [28], induced local morphology and wettability changes may bring failure of dielectric properties and electrofluidic device dynamics. A safe design for multilayer insulator should be such that the electric field is distributed uniformly. But, many characteristics, like leakage current [29], contact angle change [23] and scanning Kelvin probe [30], are measured to monitor and analyse degradation in the EWOD system. However, in [23,29,30], the nature of the failure modes for single-layer insulator and multilayer insulator was not clearly discussed. This motivated us to investigate this further. Considering the complexity of the two-phase electrofluidic device (like electrofluidic display, EFD), it makes sense to identify and distinguish the various failure modes that occur. In addition, we propose how to track the failure modes as a function of ageing time, which opens the door to quantify device performance and screen for quality in manufacturing. To the best of our knowledge, such a study in the context of EFD has not been performed at all. Thermal ageing is a method widely used to accelerate reliability testing in the display industry, which has also been introduced for the reliability evaluation of an electrowetting-based system [31]. It is generally assumed that thermal ageing provides an efficient way to accelerate the failure of devices. Furthermore, accelerated lifetime testing provides the opportunity to identify and analyse the nature of the various failure modes.

In this work, we proposed a detailed dielectric safe designed mutilayer insulator (P/AF) based on a simple bilayer model for a planar dielectric in series. This bilayer dielectric will be compared with a single layer (AFX) to explore the specific advantages and disadvantages of the multilayer approach. Particular attention will be paid to different failure modes related to dielectric failure and switching failure. To this end, accelerated thermal ageing was carried out on the devices by monitoring the leakage current and capacitance. The performance degradation has been monitored as a function of ageing time. We have aimed to establish a relationship between leakage current and dielectric failure, capacitance and switching performance based on the test results. Through quantitative analysis, the different switching failure modes related to no-opening and no-closing have been clearly identified. To reveal the mechanism behind the different failure characteristics, scanning electron microscopy (SEM) observation was conducted on the aged layers. Based on the obtained results, we propose an explanation for the variation of leakage current with ageing time.

# 2. Experimental

## 2.1. Bilayer insulator design

Using a dielectric with high dielectric constant in combination with an insulator which has a low dielectric constant leads to larger voltage drops on the layer with low constant, causing the breakdown of the latter. ParyleneC and Teflon AF1600X have similar dielectric constant ($\varepsilon_r = 3.15$ for ParyleneC and $\varepsilon_r = 1.934$ for AFX) [25], which promotes a relatively uniform electric field across the

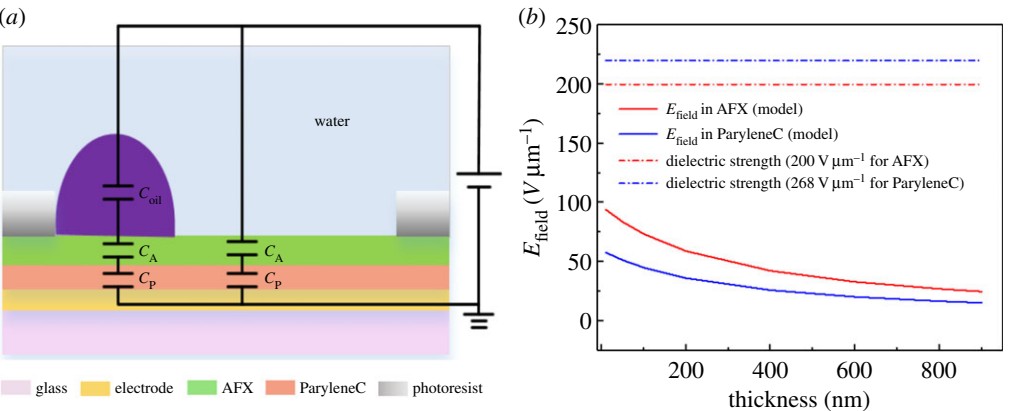

**Figure 1.** The theoretical model for mutilayer safe thickness region based on electric field distribution. (*a*) A simple series capacitance model of EFD system. (*b*) The electric field of different layer versus dielectric thickness. The theoretical breakdown strength of ParyleneC and AFX is also shown.

layers. According to the equivalent capacitance diagram (shown in figure 1*a*), when oil conductivity is ignored, the electric field strength across the fluoropolymer and ParyleneC layers can be given by

$$E_P = \frac{C_P}{C_P + C_A} \frac{V_0}{d_P} = \frac{\varepsilon_P \, d_A}{\varepsilon_P \, d_A + \varepsilon_A \, d_P} \frac{V_0}{d_P} \tag{2.1}$$

and

$$E_A = \frac{C_A}{C_P + C_A} \frac{V_0}{d_P} = \frac{\varepsilon_A \, d_P}{\varepsilon_P \, d_A + \varepsilon_A \, d_P} \frac{V_0}{d_P}, \tag{2.2}$$

where $C$ is the capacitance, $V$ the voltage, $V_0$ the applied voltage, $d$ the thickness and the subscripts $P$ and $A$ denote the ParyleneC layer and AF1600X film of the platform in device, respectively. At a fixed voltage of 30 V, and a thickness of ParyleneC of 503 nm, as receiver by the factory the electric field strength for each layer as a function of AFX thickness is plotted in figure 1*b*. The safe thickness design region of AFX can be obtained when the model calculation is less than the given dielectric strength (dotted line which is the dielectric strength of the two dielectric materials, $E_{\mathrm{ParyleneC}} \approx 268 \ \mathrm{V} \ \mu\mathrm{m}^{-1}$ [25], $E_{\mathrm{AFX}} \approx 200 \ \mathrm{V} \ \mu\mathrm{m}^{-1}$ [32]). As shown in figure 1*b*, the two layers always stay in a safe region, and a voltage of 30 V is applied. The thickness of AFX is chosen to be about 800 nm because the electrical field in two layers is similar, thus minimizing electric field gradient in the contact area.

## 2.2. Fabrication of electrofluidic display devices

The bilayer insulator was composed of a ParyleneC layer (dielectric function, 503 nm thickness) and an amorphous fluoropolymer layer (AF1600X, which carries both a hydrophobic and a dielectric function, 835 nm thickness). In this study, the ParyleneC layer was deposited on indium tin oxide (ITO) glass by vapour deposition. The amorphous fluoropolymer layer was obtained by spin-coating with 4.2 wt% AF1600X in FC43 at 1400 r.p.m. As a comparison, another sample with single-layer insulator of AF1600X of 1359 nm in thickness was prepared by spin-coating at 1000 r.p.m. The coated plates were annealed in an oven at 185°C for 30 min. A reactive ion etching with 5 W power was conducted to improve the wettability of AF1600X surface for further photoresist (PR) coating. The pixel array was constructed by lithography process with PR (HN-018N). Coloured oil (0.21 M) dissolved in decane ($C_{10}H_{22}$, $\varepsilon_r \sim 2.2$) was dosed in polar liquid (water in our case). The filled cells were edge-sealed by a pressure-sensitive adhesive attached to the cover plate.

## 2.3. Reliability measurements and analysis

A thermal ageing test was conducted to investigate the reliability of the device in long-time operation. The ageing was implemented in a test chamber (laboratory furnace) with temperature and humidity controller. The EFD devices were stored in the chamber with settings of 50°C and 55% humidity. To determine the failure modes of the tested devices, the DC voltage was applied by a DC power

rsos.royalsocietypublishing.org    R. Soc. open sci. **5**: 181121

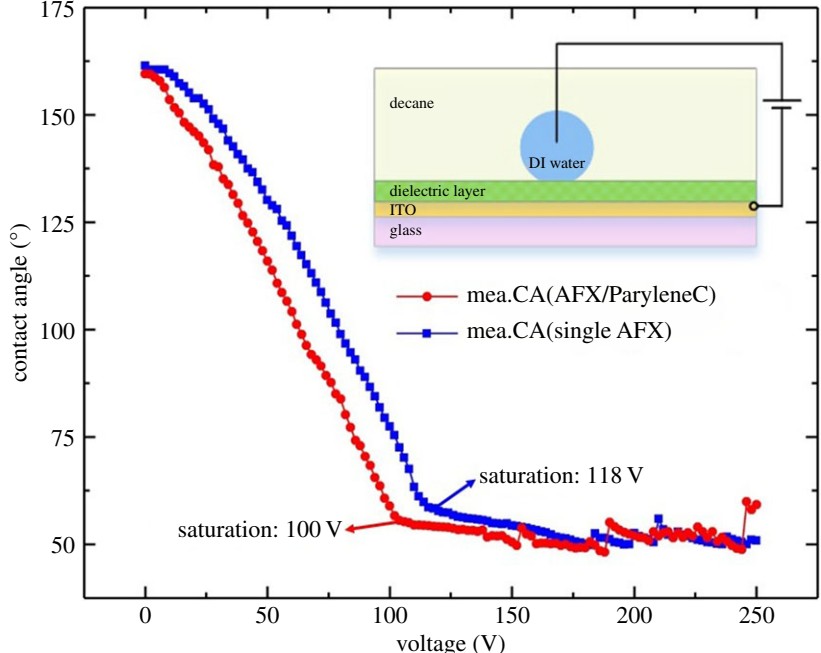

**Figure 2.** Electrowetting profiles for two-phase fluid open system of oil/water on single AFX (1359 nm) and ParyleneC/AFX (AFX: 835 nm, ParyleneC: 503 nm) coatings. The applied voltage was increased in steps of 2 V s$^{-1}$ from 0 to 250 V.

supply (PPS3003S, GAATTEN) and electrical relay, with an amplitude of 30 V square waveform. In the tests, DC voltage was applied for leakage current measurement (6482, Keithly) and capacitance measurement with a precision LCR meter (TH2828, TongHui Electronics) with LabVIEW programmes as the controlling software. In each measurement, the applied voltage was increased and then decreased in steps of 0.5 V s$^{-1}$ between 0 and 30 V for one cycle, and the voltage was never turned off between the voltages step. As one of our design rules, the EFD devices should always be working at the ideal Young–Lippmann region which is far below the threshold voltage of EW saturation. To confirm this, the electrowetting performance for a two-phase fluid open system of oil/water (as shown in figure 2, we placed a 10 µl deionized (DI) water droplet in decane) on the single AFX and ParyleneC/AFX was measured. It is clear that the driving voltage (30 V) of EFD is still far away from the EW saturation voltage of each dielectric configuration (as shown in figure 2, the saturation voltages of single AFX and ParyleneC/AFX are around 118 V and 100 V, respectively), which means that the EFD devices were all working in the linear Young–Lippmann region [30].

# 3. Results and discussion

## 3.1. Failure modes of insulators with ageing

The current response to the ageing time by stepwise voltage for different devices is shown in figure 3. Upon increasing the voltage, the oil ruptured gradually, then the fluoropolymer surface got in contact with the water, resulting in the change of the current. It should be noted that the two devices have similar dielectric thickness. The current response of bilayer device before the ageing test (0 V red curve) was much higher; however, the current decreased significantly after the first ageing test. For the subsequent ageing times, the current response stabilized (figure 3a). By contrast, the current response of the single AFX device sharply increased with ageing time (figure 3b). For an applied voltage of 30 V, the current of bilayer (P/A) device was about 100 nA cm$^{-2}$ higher than that of single AFX device before the ageing test. However, the current of the bilayer device exhibited a stable value which was much lower than that of a single-layer device after the first ageing test.

To explain these results, we envisage that while the voltage is kept constant, charges will get trapped predominantly on the interface of Teflon and ParyleneC. As the Teflon AFX was inherently porous, the charges would penetrate it when a voltage was applied. However, the impenetrable ParyleneC blocked

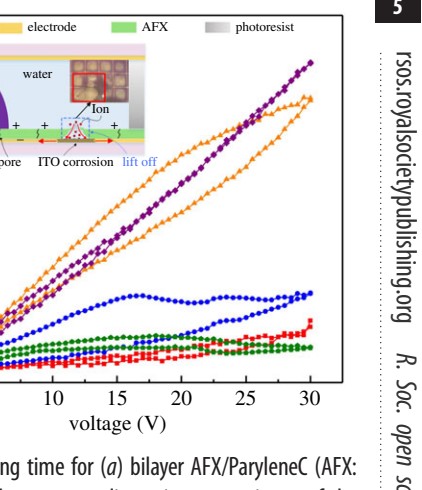

**Figure 3.** Leakage current measurements with stepwise applied voltage at different ageing time for (*a*) bilayer AFX/ParyleneC (AFX: 835 nm, ParyleneC: 503 nm) device. In the inset, the corrosion process is depicted and the corresponding microscopy picture of the damage is given. (*b*) Single AFX (AFX: 1359 nm) device. Diagrams and micrographs of ITO corrosion in different structure. In the inset, the corrosion process is depicted and the corresponding microscopy picture of the damage is given.

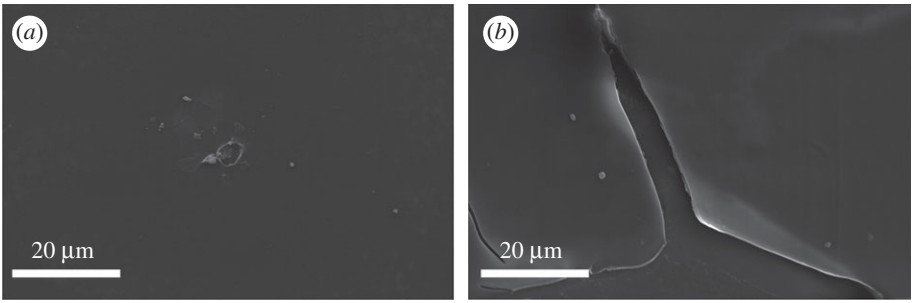

**Figure 4.** SEM images of the dielectric aged 96 h. (*a*) Bilayer (P/A) showing the damage of the AFX layer with a small hole and (*b*) single layer (AFX) showing the large area detachment of the AFX from ITO glass.

these charges and were therefore prevented from reaching the ITO. During the application of the voltage, the charge trapped into dielectric layer due to the pores or defects leads to a current, creating a flux of aqueous $OH^-$ towards the anode. The removal of electrons sustains the redox reaction between conducting liquid and ITO electrode, the overall electrochemical reaction is assumed to be $2In_2O_3 \rightarrow 4In^{3+} + 3O_2(g) + 12e^-$, and the reaction of the tin content in the ITO during anodization is as follows: $SnO_2 \rightarrow Sn^{4+} + O_2 + 4e^-$, $Sn^{4+} + 4OH^- \rightarrow Sn(OH)_4$, which leads to ITO corrosion. Some crystallites are formed and increased on the surface of the ITO $SnO_2$; they cover and increase their proportion of the surface which leads to decreasing current [33]. In this process, the contact area between electrolyte and ITO electrode is an essential element determining the electrochemical reaction rate and influencing the current.

For the bilayer structure, ParyleneC acted to cover practically all pin-holes in the AFX layer. As a result, the ions and water are blocked from reaching the ITO electrode, resulting in a much lower degree of corrosion. This is reflected by the decreasing current with ageing time. As shown in figure 4*a*, the surface morphology has been observed through SEM (magnified 8000×). It could be seen that the AFX layer was broken to form a hole, the ParyleneC layer had been damaged a little bit, while a small part of ITO corrosion could be detected. The SEM image further demonstrates the electrical current variation of the bilayer device. On the contrary, for the single AFX structure, the corrosion continued practically indefinitely because the ageing treatment resulted in AFX detachment. The detachment of the dielectric layer allowed adsorption of ions as well as invasion of liquid [34] into ITO electrode. Preferential anion adsorption has even been experimentally validated for electrowetting on fluoropolymers, which was attributed to the increase in current [24]. As confirmed by the SEM image shown in figure 4*b*, the AFX layer was completely detached from the underlying ITO glass, and the electrolyte encountered ITO glass by diffusion causing large area ITO corrosion.

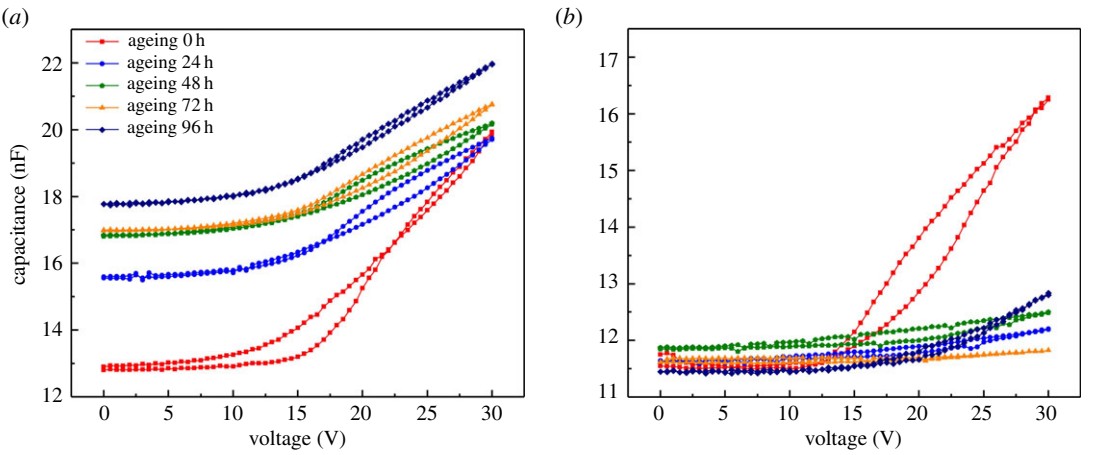

**Figure 5.** Capacitance measurements on stepwise applied voltage at different ageing time for (*a*) bilayer (P/A) device (AFX: 835 nm, ParyleneC: 503 nm) and (*b*) single AFX device (AFX: 1359 nm).

## 3.2. Switching behaviour failure with ageing

The capacitance responses to the ageing time by stepwise voltage for bilayer (P/A) device and single AFX device are shown in figure 5*a,b*. It was observed experimentally that the devices operated well (an apparent inflection point can be observed at threshold voltage in 0 h red curve) before ageing. The threshold voltage of bilayer (P/A) device was about 16 V, while that of the single AFX device was about 18 V. Upon increasing ageing time, the capacitance curve became flatter, reflecting that more and more pixels were damaged. In the absence of an electric field, the oil film always lies on the hydrophobic layer stably as the illustration shown in figure 6*a*, the capacitance referred to the total capacitance of oil film and dielectric films. The minimum capacitance (0 V) between the two devices as a function of ageing time is shown in figure 6*a*. It illustrates that more and more pixels could not be closed in the bilayer (P/A) device which could result in the increase of capacitance over ageing time. The capacitance of the single AFX device was stable under ageing, which implies that the pixels in the single AFX device were closing well even after ageing.

When the voltage is gradually increased, the fluoropolymer becomes more hydrophilic and water contact becomes favourable, which results in the oil film being pushed aside. As a result (see the illustration shown in figure 6*b*), the capacitance increases gradually upon increasing the applied voltage. By plotting the capacitance at 30 V as a function of ageing time, the result reflected that the pixels of the bilayer device could open well even after ageing (figure 6*b*). But, the pixels could not open well in single AFX device after ageing. The capacitance decreased sharply at ageing time of 24 h (figure 6*b*). The number of opened pixels in the bilayer device is always higher than in the single AFX device. To explore the relationship between switching behaviour and dielectric failure of devices, the following quantities are introduced: $\Delta C_t / \Delta C_0$, where $\Delta C_t$ is the difference between the capacitance at 30 V and the capacitance at 0 V for a given ageing time $t$, and $\Delta C_0$ is the capacitance difference between 30 and 0 V before ageing. The assumption is that $\Delta C_t$ is proportional to the number of pixels that open at 30 V at an ageing time $t$. (If no pixels open the capacity, it is not expected to change with applied voltage.) The ratio of $\Delta C_t$ and $\Delta C_0$ is therefore the relative decrease of opening pixels as a function of time. $\Delta I_t$ is the difference between the current at 30 V and the current at 0 V for a given ageing time $t$, and $\Delta I_0$ is the current difference between 30 and 0 V before ageing. The assumption is that $\Delta I_t$ is related to the dielectric failure extent at an ageing time $t$. (If there is no failure on the dielectric, the current is not expected to change with applied voltage.) The ratio of $\Delta I_t$ and $\Delta I_0$ is therefore the extent of dielectric failure as a function of time. The results show that the ratio of $\Delta C_t / \Delta C_0$ of a single AFX device sharply increases after 72 h ageing. At the same time, a decrease in $\Delta I_t / \Delta I_0$ has been shown for single AFX, suggesting that the dielectric failures have a strong connection with the switching failure. We will seek to validate this hypothesis further in future work. According to the comparison of $\Delta I_t / \Delta I_0$, it is found that the dielectric failure in a single AFX was more serious than that in the bilayer device.

The threshold voltage is the minimum voltage required to rupture the oil film, resulting in a rise in capacity (inset of figure 6*d*). With a linear fit of the measured capacitance curve above the threshold voltage, we can obtain a slope. The physical significance of this slope is related to the response uniformity in the pixels. So, if pixels do not open, the slope will decrease sharply. Therefore, we take

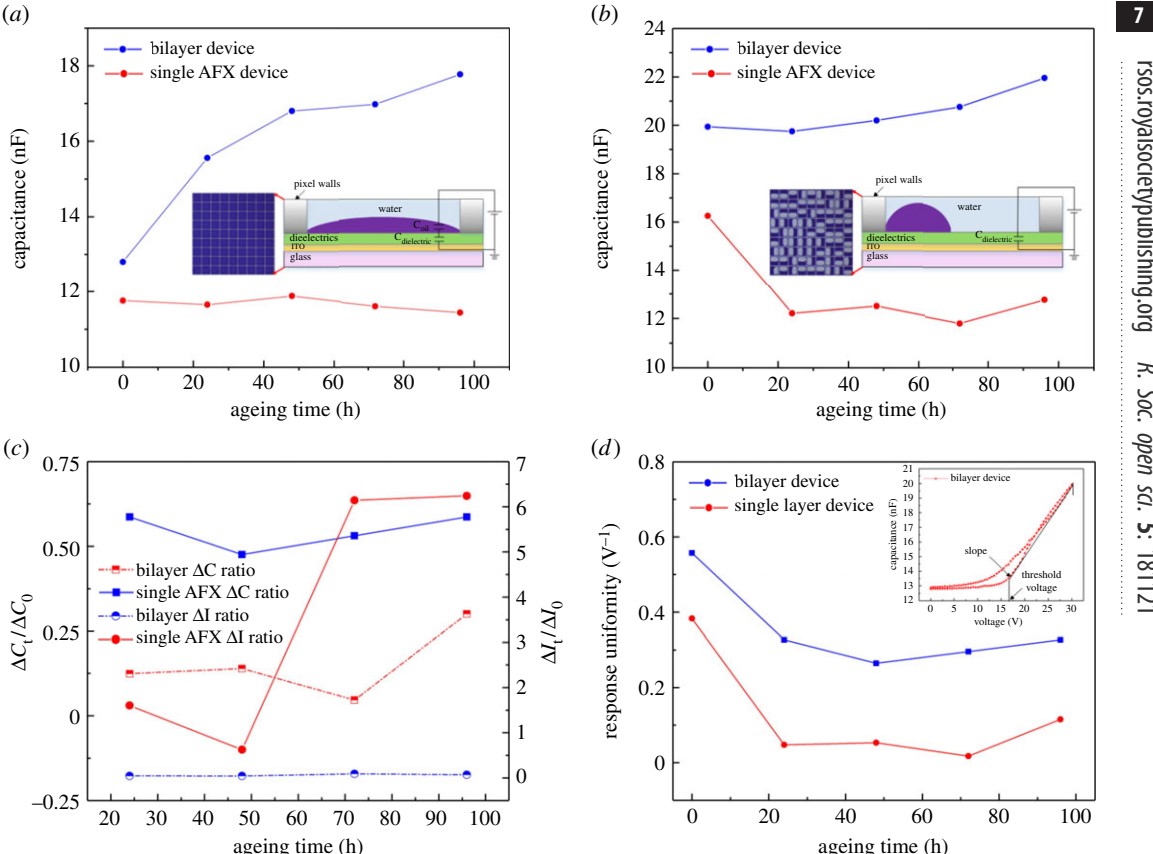

**Figure 6.** The different switching behaviour failure mode analysis: (*a*) the minimum capacitance comparison between bilayer (P/A) device and single AFX device in the absence of voltage (0 V), (*b*) the maximum capacitance comparison between bilayer (P/A) device and single AFX device when a voltage was applied at 30 V, (*c*) the comparison of $\Delta C_t/\Delta C_0$ and $\Delta I_t/\Delta I_0$ between bilayer (P/A) device and single AFX device over ageing time and (*d*) the slope obtained by linearly fitting for the comparison of response uniformity over ageing time.

the value of the slope (in $V^{-1}$) to reflect the response uniformity. The response uniformity as a function of ageing time is plotted in figure 6*d*. It has been found that thermal ageing had significant influence on device operation during the first hours of ageing. However, the operation of bilayer (P/A) device is always better than that of single AFX device.

## 4. Conclusion

The different failure modes including dielectric failure and switching failure of electrofluidic devices were compared with bilayer (ParyleneC/AFX) and single AFX. The reliability performance was systematically studied by tracking the applied voltage-dependent leakage current and capacitance changes (I–V and C–V curves) with thermal ageing time. It has been found that ultimate failure occurred upon dielectric breakdown, leading to the direct contact between electrolyte and ITO surface, following the ITO corrosion which eventually affects the current. The corrosion area and rate has a significant influence on current variation caused by the detachment of the AFX film above ITO electrode in the single AFX device. This results in ITO corrosion in the larger area; hence, the current increases sharply. As for the bilayer device, superior adhesion between ParyleneC and electrode reduced the ITO corrosion effectively, and as such the current has been effectively decreased. According to the C–V curve, threshold voltage can be clearly identified. By comparing the minimum capacitance with maximum capacitance, it has been shown that the bilayer device was more likely to suffer from the failure mode of no-closing, while the dominant failure mode of the single AFX device was no-opening. The $\Delta C_t/\Delta C_0$ and $\Delta I_t/\Delta I_0$ data in our study did not show a clear connection between the dielectric failure and switching failure.

Data accessibility. Test data and calculation model input files are deposited in Dryad Digital Repository: http://dx.doi.org/10.5061/dryad.614j89q [35]. Instructions are provided to ensure accurate replication of the experiments presented in this paper.

Authors' contributions. B.D. carried out the experiments, participated in data analysis, prepared figures and drafted the manuscript. B.T. conceived the experiments and reviewed the drafts. H.L. prepared the ParyleneC samples for analysis. R.Z. provided support on equipment training. B.T., J.G., A.V.H. and G.Z. analysed the data and revised the manuscript. All authors gave final approval for publication.

Competing interests. We declare we have no competing interests.

Funding. This work was supported by Program for Chang Jiang Scholars and Innovative Research Teams in Universities (no. IRT_17R40), Guangdong Innovative Research Team Program (no. 2011D039), Science and Technology Project of Guangdong Province (nos. 2016A010101023 and 2016B090918083), Science and Technology Program of Guangzhou (no. 201607010203), Science and Technology Project of Shenzhen Municipal Science and Technology Innovation Committee (GQYCZZ20150721150406), Guangdong Provincial Key Laboratory of Optical Information Materials and Technology (grant no. 2017B030301007), Guangzhou Key Laboratory of Electronic Paper Displays Materials and Devices (201705030007) and MOE International Laboratory for Optical Information Technologies and the 111 Project.National Key Research and Development Program of China (no. 2016YFB0401502).

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
