## [Reviewer comments · Royal Society Open Science]

Review History

RSOS-181121.R0 (Original submission)

Review form: Reviewer 1 (Papathanasiou Athanasios G)

Is the manuscript scientifically sound in its present form?

Yes

Are the interpretations and conclusions justified by the results?

Yes

Is the language acceptable?

Yes

Is it clear how to access all supporting data?

Not Applicable

Do you have any ethical concerns with this paper?

No

Have you any concerns about statistical analyses in this paper?

I do not feel qualified to assess the statistics

Recommendation?

Accept with minor revision (please list in comments)

Comments to the Author(s)

The manuscript presents a study concerning the robustness of electrowetting based display (electrofluidic display) pixels. The robustness is studied by means of performing electrical tests, namely, I-V and C-V measurements in two dielectric configurations, a single and a multilayer one. The manuscript is well and concisely written (despite some typos in instrument names, p5 line 14, or others, please check) and offers valuable knowledge in the technology community dealing with such displays.

I would just suggest the authors to present values of the saturation voltages of each dielectric configuration in order to just have idea of how they perform their tests from the saturation. And since saturation is also connected with material degradation it should be nice to have this info.

In conclusion I suggest publication after considering the above comment.

--

Athanasios G. Papathanasiou

Assistant Professor

School of Chemical Engineering, Section II

National Technical University of Athens

Zografou Campus, 15780, Athens, Greece

Phone: +30 210 772 3234

FAX: +30 210 772 3298

web: <http://www.chemeng.ntua.gr/people/pathan>

Review form: Reviewer 2

Is the manuscript scientifically sound in its present form?

Yes

Are the interpretations and conclusions justified by the results?

Yes

Is the language acceptable?

Yes

Is it clear how to access all supporting data?

Yes

Do you have any ethical concerns with this paper?

No

Have you any concerns about statistical analyses in this paper?

No

Recommendation?

Accept with minor revision (please list in comments)

Comments to the Author(s)

The authors evaluate the reliability of EFD devices under thermal aging by examining the applied voltage dependent leakage current and capacitance changes with thermal aging time.

My comments to the authors about the manuscript is related with a few typos:

Figure 5(d) legend of bilayer device.

Caption of Figure 5(b) should be "when a voltage applied at 30V"

Page 9: "It has found that" should be "It has been found that"

Decision letter (RSOS-181121.R0)

17-Oct-2018

Dear Miss Dong

On behalf of the Editors, I am pleased to inform you that your Manuscript RSOS-181121 entitled "Failure modes analysis of electrofluidic display under thermal aging" has been accepted for publication in Royal Society Open Science subject to minor revision in accordance with the referee suggestions. Please find the referees' comments at the end of this email.

The reviewers and handling editors have recommended publication, but also suggest some minor revisions to your manuscript. Therefore, I invite you to respond to the comments and revise your manuscript.

- Ethics statement

- Data accessibility

<http://datadryad.org/submit?journalID=RSOS&manu=RSOS-181121>

- **Competing interests**

- **Authors' contributions**

- **Acknowledgements**

- **Funding statement**

Because the schedule for publication is very tight, it is a condition of publication that you submit the revised version of your manuscript before 26-Oct-2018. Please note that the revision deadline will expire at 00.00am on this date. If you do not think you will be able to meet this date please let me know immediately.

1) Identifying all the changes that have been made (for instance, in coloured highlight, in bold text, or tracked changes);

on behalf of Prof. R. Kerry Rowe (Subject Editor)
openscience@royalsociety.org

Reviewer comments to Author:

Reviewer: 1

Comments to the Author(s)

The manuscript presents a study concerning the robustness of electrowetting based display (electrofluidic display) pixels. The robustness is studied by means of performing electrical tests, namely, I-V and C-V measurements in two dielectric configurations, a single and a multilayer one. The manuscript is well and concisely written (despite some typos in instrument names, p5 line 14, or others, please check) and offers valuable knowledge in the technology community dealing with such displays.

I would just suggest the authors to present values of the saturation voltages of each dielectric configuration in order to just have idea of how they perform their tests from the saturation. And since saturation is also connected with material degradation it should be nice to have this info. In conclusion I suggest publication after considering the above comment.

--

Athanasios G. Papathanasiou
Assistant Professor
School of Chemical Engineering, Section II
National Technical University of Athens
Zografou Campus, 15780, Athens, Greece
Phone: +30 210 772 3234
FAX: +30 210 772 3298
web: <http://www.chemeng.ntua.gr/people/pathan>

Reviewer: 2

Comments to the Author(s)

The authors evaluate the reliability of EFD devices under thermal aging by examining the applied voltage dependent leakage current and capacitance changes with thermal aging time.

My comments to the authors about the manuscript is related with a few typos:

Figure 5(d) legend of bilayer device.

Caption of Figure 5(b) should be "when a voltage applied at 30V"

Page 9: "It has found that" should be "It has been found that"

Author's Response to Decision Letter for (RSOS-181121.R0)

See Appendix A.

Decision letter (RSOS-181121.R1)

30-Oct-2018

Dear Miss Dong,

I am pleased to inform you that your manuscript entitled "Failure modes analysis of electrofluidic display under thermal aging" is now accepted for publication in Royal Society Open Science.

on behalf of Prof. R. Kerry Rowe (Subject Editor)
openscience@royalsociety.org

Appendix A

Failure modes analysis of electrofluidic display under thermal aging

Response to reviewers' comments and suggestions

We thank the reviewers for their careful reading and thoughtful comments on our manuscript. We have carefully taken the comments into consideration in preparing our revision. Our point-by-point responses to the comments are outlined below. The changes are also identified by **highlighting with red color** in the revised manuscript. Corrections to typographic errors are not listed here.

List of Changes:

Number	Revised contents	Position
Change 1	We have revised the instrument names into correct expression.	Page 5,line 13-15
Change 2	We have added more detailed description about the saturation voltage of each dielectric configuration in Fig2	Page 5
Change 3	We revised the Figure 5(d) legend into correct	Page 8,line55
Change 4	We revised the sentences into correct expression.	Page 9,line 10

Reviewer: 1

Comments to the author(s)

(1) The manuscript is well and concisely written (despite some typos in instrument names, p5 line 14, or others, please check) and offers valuable knowledge in the technology community dealing with such displays.

Response:

Thank you for your careful reading of our manuscript.

Page5 line13, the DC voltage applied by a DC power supply (GAATTEN pps30037-35) **has been changed to** the DC voltage

applied by a DC power supply (PPS3003S, GAATTEN) in the revised manuscript.

page 5 line14. Thank you for your careful reading of our manuscript. “leakage current measurements (Keithly PA meter 6482) ” **has been changed to** ” “leakage current measurement (6482,Keithly) ” in the revised manuscript.

Page 5 line15. capacitance measurement Impedance Analyzer (TH2828 Precision LCR Meter) **has been changed to** capacitance measurement Precision LCR Meter (TH2828, TongHui electronics) in the revised manuscript.

(2) I would just suggest the authors to present values of the saturation voltages of each dielectric configuration in order to just have idea of how they perform their tests from the saturation. And since saturation is also connected with material degradation it should be nice to have this info.

Response:

We fully understand the concern from the reviewer. As one of our design rules, the EFD devices should always working at the ideal Young-Lippmann region which is far below the threshold voltage of EW saturation. To confirm this, the electrowetting performance for two-phase fluids open system of oil/water (as shown in Fig.2, we placed 10 μ L DI water droplet in decane) on the single AFX & ParyleneC/AFX was measured. It is clear that, the driving voltage (30V) of EFD is still far away from the EW saturation voltage of each dielectric configuration (as shown in Fig.2, the saturation voltages of single AFX and ParyleneC/AFX are around 118V and 100V respectively), which means the EFD devices were all working in the linear Young-lippmann region[30]. The above information was also added into our revised manuscript(see page6 line19).

Fig2. Electrowetting profiles for two-phase fluids open system of oil/water on Single AFX (1359nm) and ParyleneC/AFX (AFX: 835nm, ParyleneC: 503nm) coatings. The applied voltage was increased in steps of 2V/s from 0V to 250V.

Reviewer: 2

Comments to the author(s)

The authors evaluate the reliability of EFD devices under thermal aging by examining the applied voltage dependent leakage current and capacitance changes with thermal aging time.

My comments to the authors about the manuscript is related with a few typos:

(1) Figure 5(d) legend of bilayer device.

Response: Thank you for your careful reading of our manuscript. It has revised in the revised manuscript

(2) Caption of Figure 5(b) should be "when a voltage applied at 30V"

Response: "when an voltage applied at 30V" **has been changed to** "when a voltage applied at 30V"

(3) Page 9: "It has found that" should be "It has been found that"

Response: Page9 line10 "It has found that" **has been changed to** "It has **been** found that".

30 Mibus, M., Hu, X., Knospe, C., Reed, M. L., Zangari, G. 2016 Failure Modes during Low-Voltage Electrowetting. *ACS Appl Mater Interfaces*. **8**, 15767-15777. (10.1021/acsami.6b02791)